# Identification, Pathogenicity of Novel Fowl Adenovirus Serotype 4 SDJN0105 in Shandong, China and Immunoprotective Evaluation of the Newly Developed Inactivated Oil-emulsion FAdV-4 Vaccine

**DOI:** 10.3390/v11070627

**Published:** 2019-07-08

**Authors:** Kai Meng, Xiaoyuan Yuan, Jiang Yu, Yuxia Zhang, Wu Ai, Youling Wang

**Affiliations:** 1Poultry Institute, Shandong Academy of Agricultural Sciences, Shandong Provincial Key Laboratory of Poultry Diseases Diagnosis and Immunology, Poultry Breeding Engineering Technology Center of Shandong Province, Jinan 250023, China; 2Shandong Key Laboratory of Animal Disease Control and Breeding, Institute of Animal Science and Veterinary Medicine, Shandong Academy of Agricultural Sciences, Jinan 250100, China

**Keywords:** FAdV-4, hydropericardium-hepatitis syndrome, pathogenicity, host innate immune response, vaccine

## Abstract

Since mid-2015, numerous outbreaks of hydropericardium-hepatitis syndrome (HHS), which is caused by a novel fowl adenovirus serotype 4 (FAdV-4), have been reported in chickens in parts of China, thereby causing huge economic losses to the poultry industry. Thus, an effective vaccine to control the further spread of infections with this hyper-virulent FAdV-4 is imperative. In this study, we isolated a novel FAdV-4 strain SDJN0105 from a broiler farm with HHS disease in Shandong Province. Pathogenicity was evaluated by the observation of clinical symptoms, necropsy changes, and pathological tissue sections after oral and intramuscular (IM) infection of Specific pathogen free (SPF) chickens. The chickens infected by IM injection all died within three days, and chickens infected via the oculonasal route died within five days post-infection (dpi). Histopathological examination revealed that the pathology was confined to heart, liver, spleen, lung, kidney, and particularly the liver. Irrespective of the inoculation route, the highest viral DNA copy numbers were detected in the livers of infected chickens. The mRNA expression levels of IL-1β, IL-6, IL-8, IFNs, TNF-α, Mx, and OASL were significantly upregulated during the viral infection. In addition, an inactivated oil-emulsion FAdV-4 vaccine was developed. The vaccine could provide full protection for SPF chickens against a lethal dose of the FAdV-4 strain SDJN0105 and a high level of antibodies. These results improve our understanding of the innate immune responses in chickens infected with FAdV-4 and the pathogenesis of FAdV-4 caused by host factors, and the developed FAdV-4 vaccine is promising as a drug candidate for the prevention and reduction of the spread of HHS in poultry in China.

## 1. Introduction

Fowl adenovirus serotype 4 (FAdV-4), which is a non-enveloped, double-stranded linear DNA virus, plays an important role in the etiology of the disease in chickens called hydropericardium-hepatitis syndrome (HHS) [1,2,3]. HHS was first reported in Angara Goth, Pakistan in 1987, and thus is also known as ‘Angara Disease’ [4,5]. HHS is a severe disease that can cause 30–80% mortality in chickens aged 3–6 weeks [6]. The main feature in HHS-affected chickens was the accumulation of transparent or straw-colored liquid in the pericardial sac and damage to the heart. Furthermore, damage has also been observed in the lungs, livers, and kidneys. Histological studies have revealed that multifocal lesions necrosis and mononuclear cell infiltration often occur in the liver of the infected chickens [7,8,9]. Since 2015, numerous major outbreaks of HHS with a high mortality rate of 30–90%, caused by a novel hyper-virulent FAdV-4, have occurred in commercial chicken farms in most parts of China, including Shandong, Henan, Jiangsu, Anhui, Hubei, and Jiangxi [10], causing a huge economic loss to the chicken industry [11].

The family of adenoviruses (AdVs) is divided into five genera: *Aviadenovirus, Mastadenovirus, Siadenovirus, Atadenovirus,* and *Ichtadenovirus* [12]. All FAdVs belong to the avian adenovirus genus, and comprise five species: FAdV-A, FAdV-B, FAdV-C, FAdV-D, and FAdV-E, which can be further divided into 12 serotypes [13,14]. Fiber, penton, and hexon form the main viral structural proteins of FAdV [15]. The fiber plays an important role in mediating the binding between the host and the virus. Penton participates in the internalization of the virus. Hexon, which is the most abundant viral surface protein, can be used for serotyping [16,17]. Thus, phylogenetic analysis of its sequences is a commonly used genotyping method [18].

The innate immune system is the first line of defense against the invasion of various pathogenic microorganisms. Pattern-recognition receptors (PRRs) are involved in the identification and removal of pathogenic microorganisms. PRRs mainly include Toll-like receptor families (TLRs), nucleotide-binding oligomerization domain-like receptor families (NLRs), C-type lectin receptor families, and retinoic acid-inducible gene I (RIG-I)-like receptors [19,20]. PRRs such as TLR-3, TLR-7, RIG-I, and MDA5 can recognize relevant viral molecular patterns and can activate the specific signaling pathways, leading to transcription of pro-inflammatory cytokines, apoptosis, and expression of type I interferons (IFNs) [21].

Since mid-2015, a novel hyper-virulent FAdV-4 has been increasingly emerged in most parts of China [22,23]. Numerous studies have focused on its evolutionary analysis, treatment, and the establishment of testing methods [24,25]. However, the associated immune responses and pathogenicity of FAdV-4-infected chickens have not been fully studied. In this research, a specific FAdV-4 strain, SDJN0105, was isolated from natural cases of HHS in Shandong Province. We investigated the pathogenicity of FAdV-4 in SPF chickens. In addition, to elucidate the role of viral tropism and the innate immune responses in viral infection, we systematically investigated the mRNA expression levels of immune-related genes in the heart, liver, spleen, lung, and kidney and the viral titers in various tissues of infected chickens. Finally, since there is a great need for an effective vaccine and no commercial vaccine against FAdV-4 caused by the novel genotype strain has been released, we developed an inactivated oil-emulsion FAdV-4 vaccine. Its protective efficacy in SPF chickens infected with a lethal dose of the FAdV-4 strain was evaluated.

## 2. Materials and Methods

### 2.1. Virus Preparation

The specific FAdV-4 strain used in the present study (SDJN0105) was isolated from a broiler farm with HHS disease in Shandong Province by the Poultry Disease Laboratory at Shandong Academy of Agricultural Sciences. The virus was propagated in SPF embryonated chicken eggs and the titers were 10^6^ TCID_50_/mL in infected chicken embryonic fibroblasts [26].

### 2.2. Electron Microscopic Examination

The chicken liver tissues were trimmed to a size of 6–8 mm^3^. After three washes with PBS, the tissues were fixed with 3% glutaraldehyde (Takara, Japan) for pre-fixation. The tissues were then stored at 4°C for two hours and gently stirred every five minutes. Subsequently, the tissues were poured into a Petri dish and incubated for 15 min at room temperature, stored in PBS at 4 °C for 1.5 h, and washed in PBS thrice. Then, the tissues were dehydrated across an ethanol gradient, and then embedded in Epon 812 at 60 °C for 48 h. The sections (<100 nm) were cut on an ultramicrotome (Japan) using a diamond knife. The sections were transferred to 200-mesh copper grids and stained with uranyl acetate for 20 min, and with lead citrate for another 10 min. The sections were examined for FAdV particles under transmission electron microscope (Hitachi, Japan) at 80 kV.

### 2.3. PCR and Phylogenetic Tree Analysis

The viral DNA of SDJN0105 was extracted from tissue samples (heart, liver, spleen, lung, and kidney) using the Viral Genomic DNA Kit. The viral DNA extraction steps were conducted according to the manufacturer’s instructions, and then the complete sequence of hexon gene was determined by Sangon Biotech. The accession number of the obtained sequence was MN102413. For phylogenetic analysis, all FAdV strains used in this research were obtained on the NCBI database. The reference strains used in this research are shown in Table 1. The phylogenetic tree was constructed using MEGA 6.0.

### 2.4. Replication and Pathogenicity of FAdV-4-SDJN0105 in Chickens

Ninety 28-old-day SPF chickens were randomly divided into three groups. Group A (*n* = 30) was injected with 0.2 mL of the viral stock through the intramuscular (IM) route. Group B (*n* = 30) was infected with 0.2 mL viral stock by pointing their eyes and nose. Group C (*n* = 30) was inoculated with 0.2 mL PBS through the IM route. At 1, 2, 3, 4, and 5 days post-infection (dpi), three chickens from each group were humanely euthanized, and the target tissues (i.e., heart, liver, spleen, lung, kidney, pancreas, cecal tonsil, and bursa of Fabricius) were harvested and stored at −80 °C until further use. Cloacal swabs were also collected for viral detection at 1, 2, 3, 4, and 5 dpi.

### 2.5. Viral Titers in Infected Chickens

The viral DNA in the target tissues (i.e., heart, liver, spleen, lung, and kidney) and cloacal cotton swabs were extracted by the method described above. The RT-PCR primers of FAdV-4 used in this study are shown in Table 2. The QuantiFast SYBR Green PCR kit (Thermo Scientific, USA) was used in quantitative RT-PCR reactions performed in 20-μL PCR tubes. The amplification program was: one cycle at 95 °C for 30 s; followed by 40 cycles of denaturation at 95 °C for 5 s and extension at 60 °C for 35 s; and then a dissociation curve analysis, using three replicates per sample.

### 2.6. The Relative mRNA Expressions of the Hearts, Livers, and Spleens from the Infected Chickens

Total RNAs of the samples (i.e., heart, liver, and spleen) were extracted, treated, quality determined, stored and reverse-transcribed to cDNA according to a previous report [27]. The primers for chicken β-actin, TLR3, TLR7, MDA5, IFN-α, IFN-β, IFN-γ, TNF-α, IL-1β, IL-2, IL-4, IL-6, IL-8, Mx and OASL genes were designed by Primer 5.0 software (http://bioinfo.ut.ee/primer5). The RT-PCR method and reaction procedures were as described above. RT-PCR primers are shown in Table 2.

### 2.7. Preparation of Oil-Adjuvant Inactivated FAdV-4 Vaccine

The virus strain SDJN0105 was propagated on chicken embryo liver (CEL) cells and vaccine production. For inactivation of the virus, formaldehyde was added to the virus culture medium collected from CEL cells infected with FAdV-4 with a 0.2% concentration in the final product. The formaldehyde-inactivated antigen solution was emulsified with an oil adjuvant at a ratio of 30:70 (*w/w*). The final dose of the inactivated oil-emulsion FAdV-4 vaccine was 10^6^ TCID_50_ each time in 0.5 mL per chicken.

### 2.8. Antibody Assay

To evaluate the antibody response of the inactivated FAdV-4 vaccine, 30 14-old-day SPF chickens were randomly divided into three groups, including 10 birds in the immunized group (group A), 10 birds in the negative control group (group B), and 10 birds in the infection control group (group C). Chickens in group A were immunized intramuscularly with 0.5 mL vaccine per chicken. Chickens in group B were injected with sterile PBS and treated as unvaccinated controls. The clinical symptoms of SPF chickens were monitored, and serum samples were collected from the pterygoid vein at 0, 7, 14, and 21 days after immunization. The level of antibodies was determined by the commercially available enzyme-linked immunosorbent assay (ELISA) kit (BioChek, Netherlands) according to the manufacturer’s instructions.

### 2.9. Virus Challenge Assay

To evaluate the protective efficacy of the inactivated vaccine, the immunization group (group A) and infection control group (group C) were challenged via IM route with the FAdV-4 strain SDJN0105 (10^6^ TCID_50_/bird, 0.2 mL) on 21 days post-vaccination (dpv). Chickens in group B were kept as a negative control. The clinical signs and survival ratio of SPF chickens were recorded daily after challenge. Ten days after challenge, all surviving chickens were euthanized, and gross lesions were evaluated.

### 2.10. Statistical Analysis

The relative mRNA expressions of target genes were calculated through the 2^−ΔΔCt^ method and quantified relative to β-actin gene expression and expressed as the fold changes in gene expression. *p*-values of < 0.05 and < 0.01 were considered to be statistically significant. The results are presented at means ± standard deviations. All data were analyzed with the Student’s *t*-test through Graph Pad Prism 6.0.

### 2.11. Ethics Statement

To evaluate the pathogenicity of FAdV-4-SDJN0105 and the immune protective effects of the inactivated vaccine against SDJN0105 strain in chickens, SPF White Leghorn chickens were used. All animal experiments were under the guidance of the Shandong Academy of Agricultural Sciences Animal Care and Use Committee. The approval number for this study is SAAS20180511.

## 3. Results

### 3.1. Electron Microscopic Examination

Figure 1A,B show that the liver nuclei in the blank control group were significantly smaller than those in the experimental group, and the edge of the liver nuclei in the experimental group was irregular. The virus particles proliferated in the liver nuclei, enlarging the nucleus, squeezing chromatin aside to form inclusions under the optical microscope, and the viral particles aggregation and arranged in a lattice form in the nuclei of hepatic cells (Figure 1C,D), indicating that the liver is a facultative tissue of FAdV-C and that the viruses were synthesized and packaged in the nucleus of hepatocytes. The adenovirus size was estimated to be between 60 nm and 90 nm.

### 3.2. Phylogenetic and Genetic Analysis of the SDJN0105 Strains

The adenovirus hexon is the major capsid protein of FAdV, containing type-, group-, and subclass-specific antigenic determinants [28]. The hexon loop 1 (L1) region is the most variable region and can identify and distinguish some or all the 12 FAdV serotypes [29]. The results of the phylogenetic analysis are shown in Figure 2. In this study, the complete hexon sequence of the FAdV-4-SDJN0105 strain has been submitted to the NCBI as GenBank Accession number MN102413. Phylogenetic tree analysis showed that the FAdV strains used in this study cluster into five major groups (FAdV-A, FAdV-B, FAdV-C, FAdV-D, and FAdV-E), and the viral strain SDJN0105 belongs to the FAdV-C group, and has the closest genetic relationship to the FAdV-4 strain (KT899325) (Figure 2).

### 3.3. Clinical Symptoms and Gross Pathology

At 2 dpi, chickens infected with SDJN0105 strain by intramuscular injection (Group A) showed obvious clinical symptoms one after another including lethargy, depression, ruffling of neck feathers, and decreased in feed intake. The morbidity was 100% (Figure 3B). Chickens began to die at 3 dpi; the pericardium had an amber-colored liquid or water-like transparent exudate, liver degeneration and necrosis, and renal swelling (Figure 3D–G). The mortality was 100% at 3 dpi in Group A. In the Group B (oculonasal route), chickens began to show varying degrees of clinical symptoms at 3 dpi and died from 4 to 6 dpi. The clinical signs and pathological changes in chickens in Group B were similar to those in Group A with a morbidity of 100% and mortality of 60%. Chickens in the uninfected control group did not show any clinical symptoms and pathological damage throughout the trial.

### 3.4. Histopathological Analysis

Tissue samples (heart, liver, spleen, lung, and kidney) of chickens infected with FAdV-4-SDJN0105 were removed for further histopathological evaluation. In terms of microscopic lesions in the liver, the hepatic lobule structure was unclear, hepatocytes were swollen, and there were some large fat vacuoles, and lots of intranuclear inclusion bodies were observed in the hepatic cells, which was accompanied by severe bleeding (Figure 4G). The accumulation of mononuclear cells was observed in some areas of the heart. In addition, edema and hyperemia were also observed in the heart (Figure 4F). In the lungs, alveolar wall thickening and fibrosis with bleeding were observed (Figure 4I). Significant loss of lymphocytes and marked lymphocytic necrosis (karyorrhexis, karyon pyknosis, and karyolysis), and bleeding were observed in the spleen (Figure 4H). In the kidney, massive degeneration of the epithelium, severe lymphocytic infiltration and severe hyperemia were detected (Figure 4J). In sum, the SDJN0105 strain can cause serious histopathological damage to various tissues in chickens, particularly the liver.

### 3.5. Viral Titers in Infected Chickens

The present study investigated FAdV-4 replication in the organs (heart, liver, spleen, lung, and kidney) of chickens infected via IM and oculonasal route as well as the cloacal cotton swabs. As shown in Figure 4, the viral titers could be detected in all the tissues and cloacal cotton swabs at 1 dpi (Group A and Group B). The highest titer was observed in the liver, suggesting that it might be the target organ of the virus. Due to the different infection pathways, the replication speed of the virus was relatively slower in Group B (oculonasal route) than in Group A (IM route). In Group A, the viral titers peaked at 2 dpi and then decreased by 3 dpi in almost all detected organs except the lung and cloacal contents (Figure 5A). In Group B, most of the tissues showed low viral titers on the first day, and the viral titers rapidly increased at 2 dpi and then peaked at 3 dpi (Figure 5B). In addition, the viral titers detected in the liver were significantly higher than in the other samples (Figure 5B). No virus was detected in any samples in the uninfected controls.

### 3.6. The Relative mRNA Expressions of Hearts, Livers, and Spleens of the Infected Chickens

Figure 6 shows that the mRNA expression trends of TLR3 and TLR7 were similar in the heart, liver, and spleen and mainly expressed in the liver and spleen. Both reached its highest level (TLR3: 10.63-fold in the liver and 14.10-fold in the spleen; TLR7: 8.10-fold in the liver and 16.53-fold in the spleen) at 2 dpi (Figure 6A,B). For MDA5, the expression level in the spleen was significantly higher than in the liver and heart at 2 dpi (33.96-fold) (*p* < 0.01; Figure 6C). Also in the spleen, IL-1β mRNA expression showed a significant upregulation (90.26-fold), particularly at 2 dpi (*p* < 0.01; Figure 6D). The expression of IL-2 peaked at 2 dpi in the heart (8.47-fold), liver (6.33-fold), and spleen (6.53-fold) (*p* < 0.05; Figure 6E). However, the expression of IL-4 peaked at 1 dpi in the liver (5.63-fold) and in the spleen (8.26-fold) (*p* < 0.05; Figure 6F). The highest increase was observed for IL-6 in the spleen at 2 dpi (954.3-fold) (*p* < 0.01; Figure 6G). In the heart, liver, and spleen, the expression level of IL-8 also significantly increased compared to the control group, especially in the spleen, which showed the most extensive upregulation (217.01-fold) at 2 dpi (*p* < 0.01; Figure 6H). The expression levels of IFNs in the spleen were significantly higher than the heart and liver. Specifically, the mRNA expression levels of IFN-α, IFN-β, and IFN-γ in the spleen were upregulated 104-fold, 602.6-fold, and 188.07-fold, respectively at 2 dpi (*p* < 0.01; Figure 6I–K). The TNF-α mRNA expression was upregulated in the heart, liver, and spleen. Especially in the spleen at 2 dpi, the expression level of TNF-α was higher than in the other two organs, reaching at 71.63-fold (*p* < 0.01; Figure 6L). In addition, the expression levels of Mx and OASL were also significantly increased compared in the heart, liver, and spleen compared to the controls (between 5.17-fold and 132.33-fold, *p* < 0.01; Figure 6M,N). These results showed that chickens infected with FAdV-4-SDJN0105 have significantly upregulated IL-1β, IL-6, IL-8, IFNs, TNF-α, Mx, and OASL, and these immune-related factors might play an important role in the innate immune response caused by viral infection.

### 3.7. Antibody Responses of Vaccinated Chickens

The specific antibody responses induced by immunizations were measured by a commercial ELISA kit. Specifically, chickens in group B showed negative results during the experiment, while chickens in group A showed positive results and the antibody titers were significantly higher than group B (*P* < 0.01). Figure 7 shows that the magnitude of the antibody response was time-dependent, and the antibody titers at 14 days post immunization (dpi) were significantly higher than at 7 dpi. In addition, the antibody titers at 21 dpi were slightly lower than those at 14 dpi.

### 3.8. Protective Efficacy of Inactivated Vaccine

In general, results showed that the inactivated vaccine can provide 100% protection against the hyper-virulent FAdV-4 strain SDJN0105. No SPF chickens died in the immunized group A and the negative control group B, whilst 8/10 of the chickens in the infection control group C died between 3 to 6 dpi. Through clinical necropsy results, no gross lesions were observed in the immunized group A (Figure 8A). By contrast, the liver of the dead chickens in the infection control group C showed swelling with necrotic foci, and flabby hearts with severe hydropericardium (Figure 8B). No gross lesions and histological changes were observed in the uninfected control (Figure 8C).

## 4. Discussion

In the present study, the FAdV-4 strain (SDJN0105) used in this research was isolated from a broiler farm with HHS disease in Shandong Province. To gain a deeper understanding of the pathogenicity of FAdV-4 to SPF chickens under different infection routes, we chose the intramuscular (IM) and oral routes for infection. The pathogenicity evaluation of SDJN0105 strain (FAdV-4 species C) indicated that SDJN0105 was a particularly virulent strain with high mortality to 28-day-old SPF chickens. However, whether SDJN0105 is highly pathogenic to chickens of different ages and even ducks remains unclear.

In this study, chickens infected with SDJN0105 via the IM route typically exhibited clinical symptoms and histopathological damage with high mortality (100%, all chickens died within 3 dpi), compared to the chickens that were infected with the virus by the oculonasal route (60%). The results indicated that the pathological damage occurred in the heart, spleen, lungs, and kidneys, and most notably in the liver. Previous research reported that FAdVs can replicate in multiple organs and produce the largest number of copies of the virus in the liver, which were consistent with our study [30]. Using electron microscopy, we could clearly see that many viral particles accumulated in the nuclei of hepatic cells, which might explain the formation of multifocal necrosis.

To understand the innate immune response caused by FAdV-4-SDJN0105 infection, the expression levels of related PRRs, proinflammatory cytokines, and IFN-stimulated genes were measured for detection and analysis in this study. The most intuitive changes were the significant up-regulation of IL-1β, IL-6, IL-8, IFNs, TNF-α, Mx, and OASL. These signaling molecules are thought to play an important role in the induction and regulation of the inflammatory response. For example, IL-1β is involved in the acute phase of inflammation, and its expression increases in tissues following bacterial or virus infections in chickens [31]. IL-6 is a multifunctional cytokine produced by a variety of cell types and has many activities, including the monocyte proliferation, acute-phase responses, neutrophil recruitment to the sites of inflammation, and immune regulation [32,33]. IL-8 is one of the most chemotactic chemokines and plays an important role in innate and acquired immunity. The main biological activity of IL-8 is to attract and activate neutrophils. The neutrophils undergo morphological changes after contact with IL-8 and migrate to the reaction site and release a series of active products; these effects can lead to local inflammatory reactions and achieve the purpose of sterilization and cell damage. In addition, IL-8 plays a role in eosinophils, basophils, and lymphocytes [34]. TNF-α also regulates complex inflammatory and immune response during viral infection, including activation and aggregation of cytotoxic T lymphocytes, neutrophils, natural killer cells, and macrophages [35,36]. The highly up-regulated expression of major proinflammatory cytokines, particularly IL-6, IL-8, IFNs, and TNF-α, can cause severe inflammatory responses in infected tissues and ultimately lead to death. A previous study reported that a significant increase in inflammatory cytokines is positively associated with tissue damage, which is consistent with our study [37].

Commercially available vaccine products for FAdV-4 are still missing in the market. Therefore, an inactivated oil-emulsion FAdV-4 vaccine formulated was developed in this study, and the antibody response and protective efficacy were also evaluated by challenging SPF chickens with the hyper-virulent FAdV-4 strain SDJN0105. No distinct clinical signs and pathological damage were observed in the immunized group, indicating that the developed vaccine was safe and could provide sufficient protection for chickens. In our study, the FAdV-4 specific antibody titers of the newly developed inactivated vaccine could reach at 6,300 in 14 days post single immunization, while the inactivated vaccine Pan et al. [38] developed had an antibody titer of less than 4,000 in 14 days post single immunization, which indicated that the newly established inactivated vaccine could produce a higher level of antibodies. In addition, Kim et al. [30] also developed an inactivated oil-emulsion fowl Adenovirus serotype 4 vaccine. Although the vaccine could provide broad cross-protection against various serotypes of fowl Adenovirus, the protection rate against FAdV-4 was only 80%. It was reported that the novel FAdV-4 has the specificity of continuous infection and cross-transmission between chickens and ducks [22,27]. It is not known whether the FAdV-4 strain SDJN0105 has pathogenicity to ducks. Similarly, it is not known whether the established vaccine has a cross-protective effect on other serotypes of FAdV. These issues will continue to be discussed in our follow-up research.

In summary, our research suggests that the FAdV-4 strain SDJN0105 can cause HHS and lead to rapid, severe, and massive death in populations of SPF chickens. After a series of studies and analyses, we speculate that the target organ of the virus is the liver and the innate immune-related genes (IL-1β, IL-6, IL-8, IFNs, and TNF-α) may play an important role in the immune response. In addition, an inactivated novel FAdV-4 vaccine was also developed, which provided complete protection against the HHS and may be potentially utilized as a drug candidate for the prevention and the control of HHS in poultry in China.

## Figures and Tables

**Figure 1 viruses-11-00627-f001:**
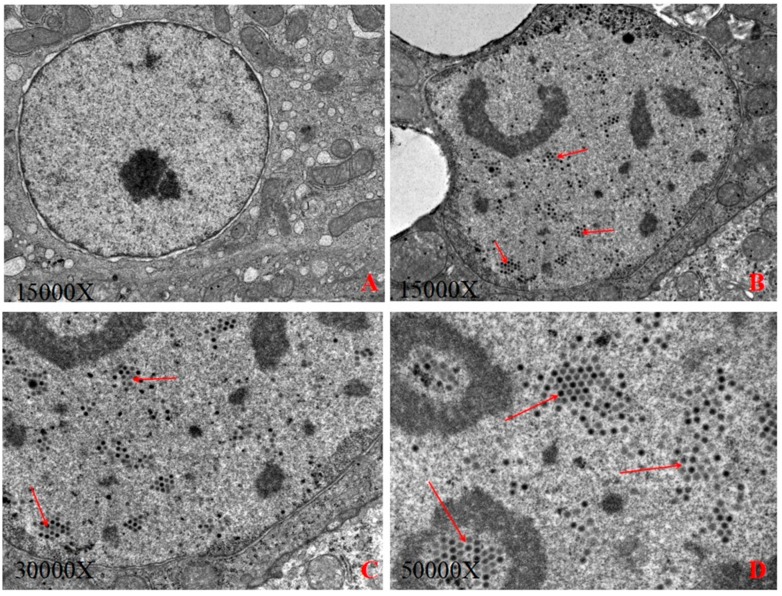
(**A**): The control group; (**B**): The virion of FAdV-4-SDJN0105 at 15,000× magnification; (**C**): The virion of FAdV-4-SDJN0105 at 30,000× magnification; (**D**): The virion of FAdV-4-SDJN0105 at 50,000× magnification.

**Figure 2 viruses-11-00627-f002:**
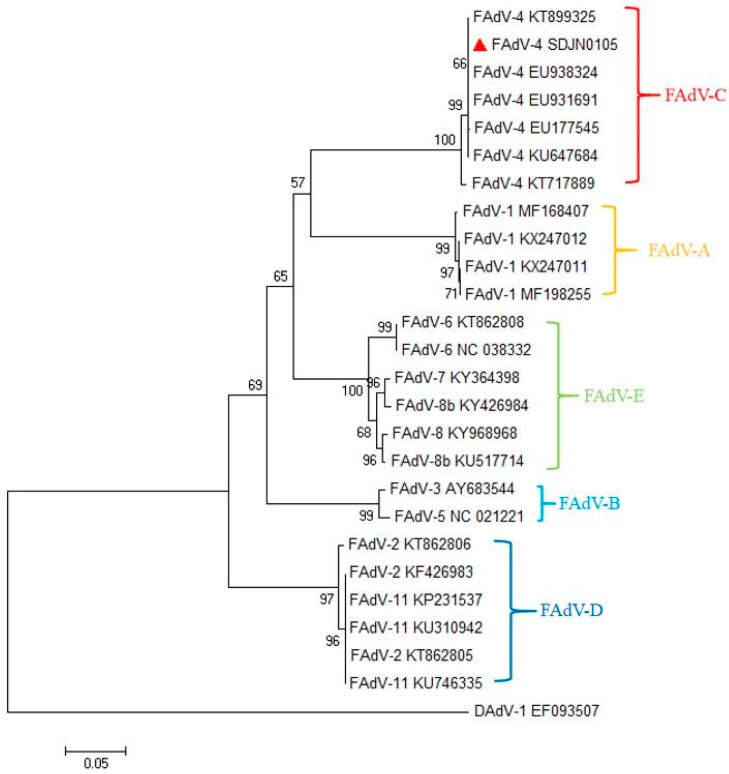
Phylogenetic tree of the isolated FAdV-4 strain (SDJN0105) based on the L1 loop of hexon gene sequences. The SDJN0105 sequenced in this figure is marked with a red triangle; other FAdV strains and DAdV strain used in this study were found on the National Center for Biotechnology website. GenBank numbers are shown in Table 2. This phylogenetic tree was generated using MEGA 6.0 software employing the neighbor-joining method and a 1000-bootstrap analysis. The scale bar is 0.05.

**Figure 3 viruses-11-00627-f003:**
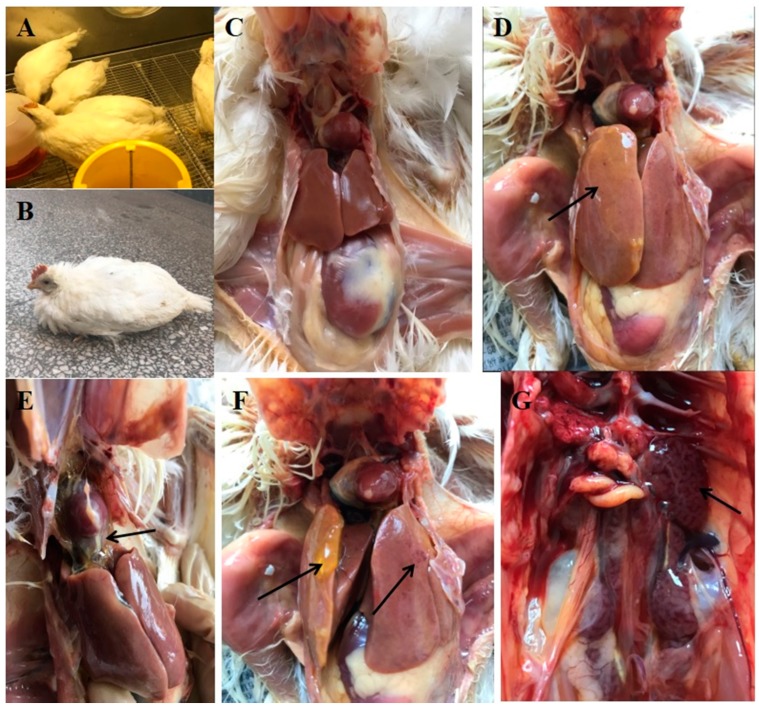
Clinical symptoms and gross pathology of FAdV-4 SDJN0105-infected chickens and uninfected controls. (**A**,**C**) Healthy chicken from the control group. (**B**) Lethargy, depression, and ruffling of neck feathers as seen in an SDJN0105-infected chicken. (**D**,**F**) Yellow jelly or watery-like exudate in the liver and liver degeneration and necrosis. (**E**) Amber colored liquid or water-like transparent exudate in the pericardial sac. (**G**) Renal hemorrhage and swelling.

**Figure 4 viruses-11-00627-f004:**
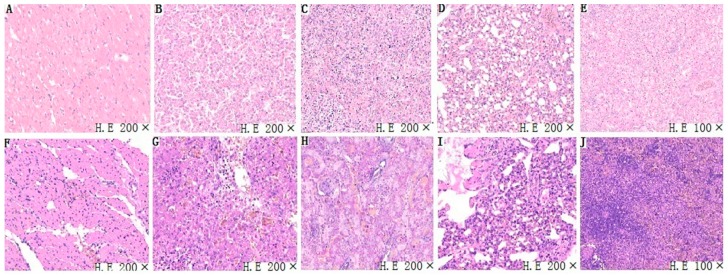
Representative light microscopy sections of tissues (i.e, heart, liver, spleen, lung, kidney) of FAdV-4 SDJN0105-infected chickens. (**A**–**E**) Heart, liver, spleen, lung, and kidney, respectively, of chickens are from the control group. Normal histopathology can be observed in the control group. (**F**–**J**) Heart, liver, spleen, lung, and kidney, respectively, are from the infected chicken.

**Figure 5 viruses-11-00627-f005:**
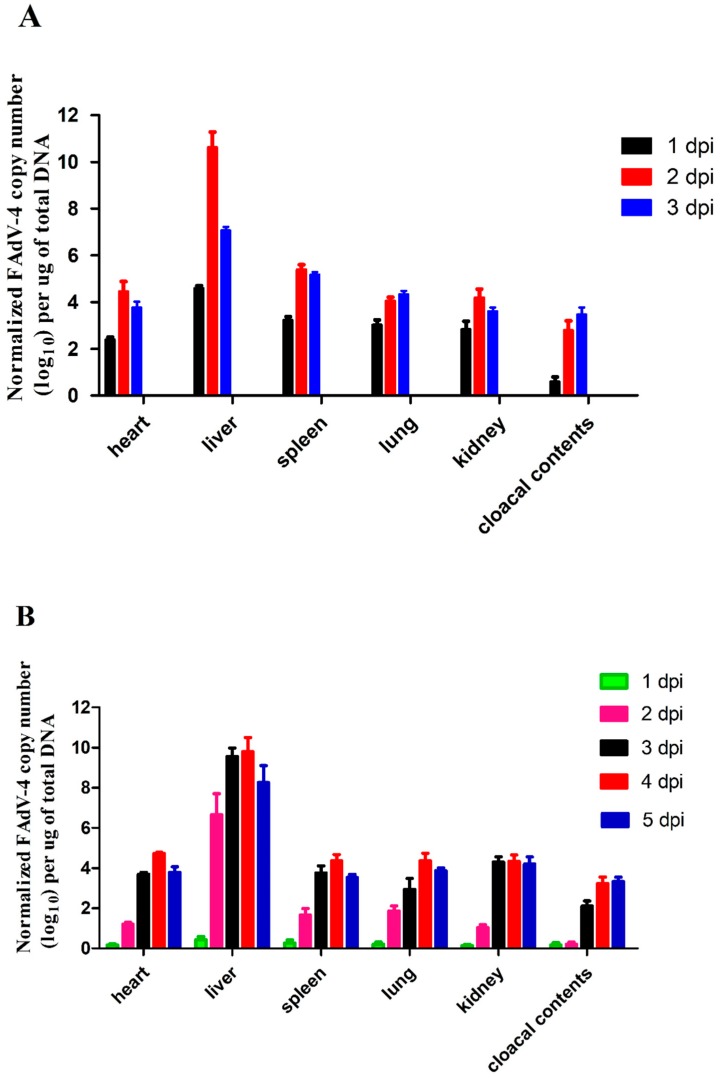
Viral titers in samples (heart, liver, spleen, lung, kidney, and cloacal contents) from FAdV-4 infected chickens. (**A**) FAdV-4-SDJN0105 DNA loads in the tissues and cloacal cotton swabs from intramuscularly infected chickens at 1, 2, and 3 dpi. (**B**) FAdV-4-SDJN0105 DNA loads in the tissues and cloacal cotton swabs from oculonasal infected chickens at 1, 2, 3, 4, and 5 dpi. Data are expressed as means ± standard deviations (*n* = 3), and each sample was analyzed in triplicate. dpi, days post-infection.

**Figure 6 viruses-11-00627-f006:**
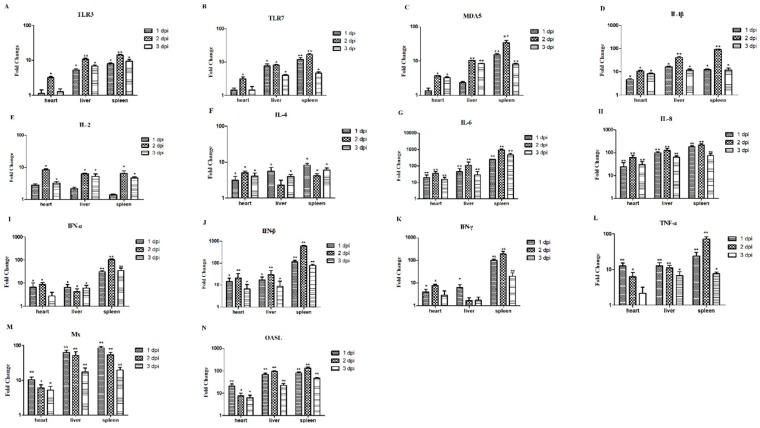
Expression profiles of immune-related genes in the hearts, livers, and spleens from IM-infected chickens. Tissue samples from chickens were collected at 1, 2, and 3 dpi. The inducible gene mRNA expression levels in the tissue samples were tested using real-time quantitative reverse transcription polymerase chain reaction (qRT-PCR) analysis. Relative expression levels were normalized to the β-actin gene and calculated using the 2^−△△Ct^ method. The Y-axis represents the fold change in target gene expression in the experimental group versus that in the control group. Data are expressed as means ± standard deviations (*n* = 3). (**A**–**N**) Expression levels of TLR3, TLR7, MDA5, IL-1β, IL-2, IL-4, IL-6, IL-8, IFN-α, IFN-β, IFN-γ, TNF-α, Mx, and OASL in the hearts, livers, and spleens of infected chickens. Differences were analyzed using the Student’s *t*-test. * *P* < 0.05; ** *P* < 0.01. dpi, days post-infection.

**Figure 7 viruses-11-00627-f007:**
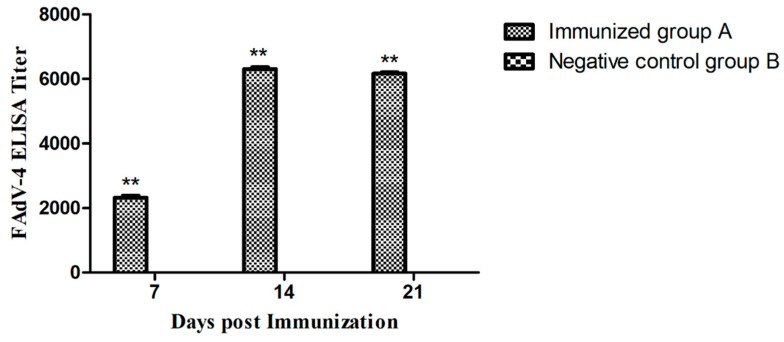
FAdV-4 specific antibody responses induced at 7, 14, and 21 days post-immunization. Chickens in the Immunized group A produced a higher antibody response than the Negative control group B (** *P* < 0.01).

**Figure 8 viruses-11-00627-f008:**
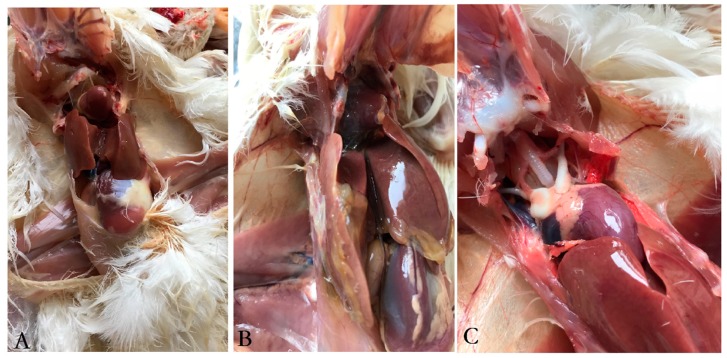
Gross lesions in the liver and heart from chicken challenged with the FAdV-4 strain SDJN0105. (**A**): Liver and heart tissues of challenged chicken immunized with the inactivated vaccine. (**B**): Liver and heart tissues from the infection control chicken. (**C**): Liver and heart from the uninfected control.

**Table 1 viruses-11-00627-t001:** The sequences information of FAdV reference strains used in this study.

Species	GenBank Numbers	Serotype
FAdV-A	MF168407	FAdV-1
	KX247011	FAdV-1
	KX247012	FAdV-1
	MF198255	FAdV-1
FAdV-B	AY683544	FAdV-3
	NC_021221	FAdV-5
FAdV-C	EU177545	FAdV-4
	EU931691	FAdV-4
	EU938324	FAdV-4
	KU647684	FAdV-4
	KT899325	FAdV-4
	KT717889	FAdV-10
FAdV-D	KT862805	FAdV-2
	KT862806	FAdV-2
	KF426983	FAdV-2
	KU746335	FAdV-11
	KU310942	FAdV-11
	KP231537	FAdV-11
FAdV-E	NC_038332	FAdV-6
	KT862808	FAdV-6
	KY364398	FAdV-7
	KY968968	FAdV-8
	KU517714	FAdV-8b
	KY426984	FAdV-8b
DAdV	EF093507	DAdV-1

**Table 2 viruses-11-00627-t002:** Primers used in this study.

Primer Name	Sequence of Oligonucleotide (5′–3′)	Size (bp)
q-FAdV-4-F	CGTCCCGACTACCCTAGAGA	150
q-FAdV-4-R	AGGTCCCGCAACTGAGACT	
β-actin-F	ATGAAGCCCAGAGCAAAAGA	120
β-actin-R	GGGGTGTTGAAGGTCTCAAA	
TLR3-F	ACAATGGCAGATTGTAGTCACCT	123
TLR3-R	GCACAATCCTGGTTTCAGTTTAG	
TLR7-F	TGTGATGTGGAAGCCTTTGA	185
TLR7-R	ATTATCTTTGGGCCCCAGTC	
MDA5-F	TGAAAGCCTTGCAGATGACTTA	120
MDA5-R	GCTGTTTCAAATCCTCCGTTAC	
TNF-α-F	CCGCCCAGTTCAGATGAGTT	130
TNF-α-R	GCAACAACCAGCTATGCACC	
IFN-α-F	CACCTTCCTCCAAGACAACGATT	108
IFN-α-R	GTGCGAGTGATAAATGTGAGGTTG	
IFN-β-F	CCTCAACCAGATCCAGCATT	180
IFN-β-R	GGATGAGGCTGTGAGAGGAG	
IFN-γ-F	TGAGCCAGATTGTTTCGATG	186
IFN-γ-R	CTTGGCCAGGTCCATGATA	
IL-1β-F	CCAGAAAGTGAGGCTCAACATTG	114
IL-1β-R	GACATGTAGAGCTTGTAGCCCTT	
IL-2-F	TACCTGGGAGAAGTGGTTACTCT	117
IL-2-R	TAGACCCGTAAGACTCTTGAGGT	
IL-4-F	AGAGCATCCGGATAGTGAATGAC	142
IL-4-R	AGATCGATACCAGTCTGAGCAAC	
IL-6-F	CCTGTTCGCCTTTCAGACCT	171
IL-6-R	GGGATGACCACTTCATCGGG	
IL-8-F	ATTCAAGATGTGAAGCTGAC	196
IL-8-R	AGGATCTGCAATTAACATGAGG	
Mx-F	CAGCTCCAGAATGCATCAGA	181
Mx-R	GGCAATTCCAGGAAGATCAA	
OASL-F	GAGATGGAGGTCCTGGTGAA	152
OASL-R	CCAGCTCCTTGGTCTCGTAG

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
