# Peer review of "Identification, Pathogenicity of Novel Fowl Adenovirus Serotype 4 SDJN0105 in Shandong, China and Immunoprotective Evaluation of the Newly Developed Inactivated Oil-emulsion FAdV-4 Vaccine"

_viruses, 2019, doi:10.3390/v11070627_

Round 1
Reviewer 1 Report
Since mid-2015, numerous outbreaks of hydropericardium-hepatitis syndrome (HHS), which is caused by a novel fowl adenovirus serotype 4 (FAdV-4), have been reported in chickens in
parts of China, thereby causing huge economic losses to the poultry industry in this country.
The authors evaluated specific and effective vaccine against type 4 caused HHS. First of all authors
isolated a novel FAdV-4 strain SDJN0105 from a broiler farm with HHS disease in Shandong Province. Pathogenicity was evaluated by observation of clinical symptoms, necropsy changes, and autopsy after oral and intramuscular (IM) infection of Specific pathogen free (SPF) chickens have been performed. The chickens infected by IM injection all died within three days, and chickens infected by oral infection died within five days post-infection (dpi). Histopathological examination revealed that the lesions was confined to heart, liver, spleen, lung, liver, and kidney. Irrespective of the inoculation route, the highest viral DNA copy numbers were detected in the livers of infected chickens. The mRNA expression levels of IL-1β, IL-6, IL-8, IFNs, TNF-α, Mx, and OASL were significantly upregulated during the viral infection. In addition, this study developed an inactivated oil-emulsion FAdV-4 vaccine. The vaccine can provide full protection for SPF chickens against a lethal dose of the FAdV-4 strain SDJN0105 and a high level of antibodies. These results improve our understanding of the innate immune responses caused by chicken infection with FAdV-4 and the pathogenesis of FAdV-4 caused by host factors, and the developed FAdV-4 vaccine is extremely promising. Definitely I am for the publication above mention manuscript.
supplementary:
Concerning the adenoviruses and infection causing HHS by the type 4 of adenoviruses the research topic of this manuscript is interesting on the field of adenovirus virology. Adenovirus strains are diverse pathogens and are responsible for several diseases almost in every continent. So every single method which can led us to decrease of diseases causing by adenoviruses is valuable. In my opinion the newly developed inactivated oil-emulsion FAdV-4 vaccine is important form epidemiological point of view.
The topic is important from the epidemiological point of view. Concerning my opinion inactivated emulsion FAdV-4 vaccine is original topic.
All the procedure of investigation and preparing vaccine is crucial and well written.
The paper is well written however in some cases needs the native speaker correction
All the text is clear and easy to read.
In my opinion conclusions are consistent with evidence and arguments are presented well.
All the arguments and conclusions are written well
Author Response
Response to Reviewer 1 Comments
Point 1: The paper is well written however in some cases needs the native speaker correction.
Response 1: Thank you very much for your comments. Based on your suggestion, the revised paper has been polished by a native speaker and the places modified in the paper have been marked in red.
Reviewer 2 Report
Continuous outbreaks of hydropericardium-hepatitis syndrome in chickens in China have resulted in many publications related with pathogenicity of fowl adenovirus serotype 4, its induction of host innate immunity, and vaccine development against fowl adenovirus serotype 4. Similarly, the authors isolated one fowl adenovirus serotype 4 strain and evaluated its pathogenicity and protective efficacy of inactivated vaccine against homologous strain. The major weakness of this study is a lack of novelty in experiment designs, results, and conclusions. It is also required to improve the readability of the manuscript.
Specific comments:
Line 36: change “Fowl adenoviruses serotype 4” to “Fowl adenovirus serotype 4.”
Lines 99-100; Table 1: The size of hexon gene is 2,814 nt in length. Sequencing 500 nt of the gene would not be sufficient for the phylogenic analysis. The phylogenetic analysis and % identity of complete sequence of hexon amino acid would be a better approach.
Lines 148-149: Include the route of viral inoculation used for the challenge.
Lines 195 and 197 (mortality of chickens): Clarify the two different mortality of chickens at 3 dpi.
Line 198; line 233 (oral inoculation in group B chickens): In the method section (lines 108-109), the chickens were inoculated by using the oculonasal route.
Line 201; 238: change “blank” to “uninfected.”
Line 289: Shedding of challenge virus in chickens should be determined in evaluating protective efficacy of vaccine.
Lines 339-342: Many studies have shown the efficacy of inactivated vaccines against FAdV-4. The authors would explain how this study can be unique compared to other previous study. One study even showed that inactivated, oil-emulsified vaccine not only provided protection against FAdV-4 in vaccinated chickens, but it also provided broad cross-protection against various serotypes of FAdV in vaccinated chickens.
Lines 345-346: Please refer to this reference: Li Rong, Li Gen, Lin Jing, Han Shaojie, Hou Xiaolan, Weng Hongyu, Guo Mengjiao, Lu Zhong, Li Ning, Shang Yingli, Chai Tongjie, Wei Liangmeng. Fowl 2018. Adenovirus Serotype 4 SD0828 Infections Causes High Mortality Rate and Cytokine Levels in Specific Pathogen-Free Chickens Compared to Ducks. Frontiers in Immunology. 9: 49. DOI=10.3389/fimmu.2018.00049.
Author Response
Response to Reviewer 2 Comments
Point 1: It is also required to improve the readability of the manuscript.
Response 1: Thank you very much for your comments. Based on your suggestion, the revised paper has been polished by a native speaker and the places modified in the paper have been marked in red.
Point 2: Line 36: change “Fowl adenoviruses serotype 4” to “Fowl adenovirus serotype 4.”
Response 2: Thank you very much for your suggestion. The sentence “Fowl adenoviruses serotype 4” has been changed into “Fowl adenovirus serotype 4” in the revised paper (Line 39).
Point 3: Lines 99-100; Table 1: The size of hexon gene is 2,814 nt in length. Sequencing 500 nt of the gene would not be sufficient for the phylogenic analysis. The phylogenetic analysis and % identity of complete sequence of hexon amino acid would be a better approach.
Response 3: Thank you very much for your constructive comments. We have determined the whole sequence of the hexon gene and re-obtained the accession number of the sequence (MN102413). In addition, the reference complete sequences of hexon gene (Table 1) were also obtained to re-developed the new phylogenetic tree (Fig. 2), and the places modified in the revised manuscript have been marked in red (Line 98-99; Line 103-104; Line 179-180; Line 185-191).
Point 4: Lines 148-149: Include the route of viral inoculation used for the challenge.
Response 4: Thank you very much for your suggestion. The sentence “the immunization group (group A) and infection control group (group C) were challenged with the FAdV-4 strain SDJN0105 (106 TCID50/bird, 0.2 mL) on 21 days post-vaccination (dpv)” has been changed into “the immunization group (group A) and infection control group (group C) were challenged via IM route with the FAdV-4 strain SDJN0105 (106 TCID50/bird, 0.2 mL) on 21 days post-vaccination (dpv)”. (Line 145)
Point 5: Lines 195 and 197 (mortality of chickens): Clarify the two different mortality of chickens at 3 dpi.
Response 5: We agree with your suggestion. The sentence “The mortality was 100% at 3 dpi in Group A.” was added in the revised paper. (Line 197). We think that the original sentence “The clinical signs and pathological changes in chickens in Group B were similar to those in Group A with a morbidity of 100% and mortality of 60%” can clarify the two different mortality of chickens. (Line 199-200)
Point 6: Line 198; line 233 (oral inoculation in group B chickens): In the method section (lines 108-109), the chickens were inoculated by using the oculonasal route.
Response 6: Thank you very much for your comments. Please forgive me for the mistake I made. We have corrected the errors in the revised manuscript. (Line 24; Line 198; Line 229; Line 233; Line 243; Line 312)
Point 7: Line 201; 238: change “blank” to “uninfected.”
Response 7: Thanks a lot. Based on your suggestion, we have changed “blank” to “uninfected” in the revised paper. (Line 201; Line 238)
Point 8: Line 289: Shedding of challenge virus in chickens should be determined in evaluating protective efficacy of vaccine.
Response 8: Thank you very much for your comments. In the early stage, the pre-assessment testing of the inactivated vaccine has been already done, and no virus was detected in the cloacal contents of 6 chickens in the vaccine group. Therefore, this part is not covered in this article.
Point 9: Lines 339-342: Many studies have shown the efficacy of inactivated vaccines against FAdV-4. The authors would explain how this study can be unique compared to other previous study. One study even showed that inactivated, oil-emulsified vaccine not only provided protection against FAdV-4 in vaccinated chickens, but it also provided broad cross-protection against various serotypes of FAdV in vaccinated chickens.
Response 9: Since mid-2015, a novel fowl adenovirus serotype 4 (FAdV-4), a high mortality rate of 30%-90%, was emerged in most parts of China. In our study, the FAdV-4 specific antibody titers of the newly developed inactivated vaccine can reach at 6,300 in 14 days post single immunization, while the inactivated vaccine Pan et al. developed has an antibody titer of less than 4,000 in 14 days post single immunization. In addition, the protection rate of the inactivated, oil-emulsified vaccine developed by Kim et al. was only 80% against FAdV-4, although it could provide broad cross-protection against various serotypes of fowl Adenovirus. The related content has been added to the discussion (Line 343-349), and two related references have been supplied to the references section of the revised paper. (Line 454-458)
Pan Q, Yang Y, Gao Y, et al. An inactivated novel genotype fowl adenovirus 4 protects chickens against the hydropericardium syndrome that recently emerged in China. Viruses, 2017, 9(8): 216.
Kim M S, Lim T H, Lee D H, et al. An inactivated oil-emulsion fowl Adenovirus serotype 4 vaccine provides broad cross-protection against various serotypes of fowl Adenovirus. Vaccine, 2014, 32(28): 3564-3568.
Point 10: Lines 345-346: Please refer to this reference: Li Rong, Li Gen, Lin Jing, Han Shaojie, Hou Xiaolan, Weng Hongyu, Guo Mengjiao, Lu Zhong, Li Ning, Shang Yingli, Chai Tongjie, Wei Liangmeng. Fowl 2018. Adenovirus Serotype 4 SD0828 Infections Causes High Mortality Rate and Cytokine Levels in Specific Pathogen-Free Chickens Compared to Ducks. Frontiers in Immunology. 9: 49. DOI=10.3389/fimmu.2018.00049.
Response 10: Thank you very much. Based on your suggestion, the reference has been added in the revised paper. (Line 348; Line 428-430)
Round 2
Reviewer 2 Report
The revised manuscript can be accepted for the publication.